# Mitochondrial Aging and Senolytic Natural Products with Protective Potential

**DOI:** 10.3390/ijms232416219

**Published:** 2022-12-19

**Authors:** Andrea Deledda, Emanuele Giordano, Fernanda Velluzzi, Giovanna Flore, Sara Franceschelli, Lorenza Speranza, Patrizio Ripari

**Affiliations:** 1Endocrinology and Obesity Unit, Department of Medical Sciences and Public Health, University of Cagliari, 09124 Cagliari, Italy; 2Independent Researcher, 90100 Palermo, Italy; 3Department of Medicine and Aging Sciences, University “G. D’Annunzio” Chieti- Pescara, 66100 Chieti, Italy; 4Department of Innovative Technologies in Medicine & Dentistry, University “G. D’Annunzio” Chieti-Pescara, 66100 Chieti, Italy

**Keywords:** phytonutrients, epigallo-catechin-gallate, oleuropein, curcumin, fisetin, quercetin, selenium, magnesium, senolytic, senescence

## Abstract

Living organisms do not disregard the laws of thermodynamics and must therefore consume energy for their survival. In this way, cellular energy exchanges, which aim above all at the production of ATP, a fundamental molecule used by the cell for its metabolisms, favor the formation of waste products that, if not properly disposed of, can contribute to cellular aging and damage. Numerous genes have been linked to aging, with some favoring it (gerontogenes) and others blocking it (longevity pathways). Animal model studies have shown that calorie restriction (CR) may promote longevity pathways, but given the difficult application of CR in humans, research is investigating the use of CR-mimetic substances capable of producing the same effect. These include some phytonutrients such as oleuropein, hydroxytyrosol, epigallo-catechin-gallate, fisetin, quercetin, and curcumin and minerals such as magnesium and selenium. Some of them also have senolytic effects, which promote the apoptosis of defective cells that accumulate over the years (senescent cells) and disrupt normal metabolism. In this article, we review the properties of these natural elements that can promote a longer and healthier life.

## 1. Introduction

Aging is largely defined as a time-dependent gradual and progressive decline in living organisms’ cellular and organ functions, leading to increased susceptibility to chronic diseases and death [1]. Senescent cells are implicated in a number of biological functions, from embryogenesis to aging. Significantly, an excessive accumulation of senescent cells is associated with a decline in regenerative capacity and chronic inflammation. Mitochondria have been increasingly identified as the pivotal players in the aging process. Mitochondrial dysfunction, characterized by a reduced oxidative capacity and a consequent increase in oxidative damage, is believed to contribute decisively and significantly to biological aging [2]. Mitochondria are the only cell organelles known to have their own DNA (mitochondrial DNA, or mtDNA) that is distinct from nuclear DNA (nDNA). They are double-membrane organelles present inside our cells, which contain the enzymes of Krebs cycle oxidative phosphorylation and are involved in the biosynthesis of fatty acids. Their main function is to produce energy through a metabolic process called oxidative phosphorylation (OXPHOS). This process occurs in the inner membrane of the mitochondria via a series of redox reactions performed by the four respiratory chain enzyme complexes [3,4]. Each mitochondrion is tailor-made to provide the needs of the cell in which it resides. Indeed, there are several mitochondria with specialized metabolic functions for many of the 250 different types of cells in our bodies [3]. A multitude of experimental studies consider the aging process to be the result of damage to cellular biomolecules due to the excessive production of highly toxic reactive oxygen species (ROS) [5]. Since mitochondria are the main producers of ROS in the cell, the mitochondrial theory of aging has been developed according to which mitochondria, with age, accumulate damage induced by excessive levels of ROS and become poorly functional or non-functional. Mitochondria are the primary source of energy in cells, and their damage has been involved in the aging process. This, over time, leads to cellular dysfunction responsible for aging and inevitable cell death. In the field of the prevention of mitochondrial damage, there is still much to understand [6]. 

Physical exercise can have a beneficial effect in counteracting mitochondrial dysfunction and favoring the biogenesis of new mitochondria [7]. Due to this action, physical exercise can be considered a non-pharmacological tool in the prevention and therapy of pathological conditions that are typical in older adults, such as cardiovascular or metabolic diseases [8,9]. As for nutrition, a dietary pattern rich in antioxidants, such as the Mediterranean diet, is known to have a protective effect on mitochondria, interacting with their dynamics [10]. In the last few years, the discovery of natural products that affect mitochondrial function has become an interesting field of research in the development of innovative drugs. Some pharmacologically active substances called senolytics, including both natural (e.g., quercetin and fisetin) and artificial substances (e.g., dasatinib, have been approved for clinical use in the United States since 2006), can stimulate apoptosis in senescent cells (SC), and have shown potential in blocking and reversing conditions associated with the elderly [11]. This review analyzes some phytochemicals and minerals involved in antioxidant and senolytic effects: oleuropein, quercetin, epigallo-catechin-gallate, fisetin, curcumin, selenium, and magnesium. 

## 2. Physiological Decline and Physiology of Aging

All organisms use metabolic processes to take advantage of energy in a very efficient way, but there is also a need to counteract the increase in entropy, which manifests as waste products and dissipative forces. This can induce chemical/physical damage in the cell [12]. Progressively, however, the entropy of an open system increases locally, and cells age, with significant alterations in their physiology. The transport of metabolic energy in biological processes takes place along hierarchical network systems that have evolved with competitive feedback mechanisms [13,14]. These networks must be able to reach any part of the body in order to provide nourishment for cells, mitochondria, and respiratory complexes and allow intracellular functions. Despite the efficiency of these systems, metabolic reactions continuously cause damage in order to take place, which ultimately has the most serious, rooted, and significant consequences at the cellular and intracellular levels and more specifically where the exchanges of nutrients and energy take place: at the level of blood capillaries and mitochondria, whose properties are not related to the size of the organism [15].These networks occupy the entire physical space and must reach the mitochondria in every area of the body, so the damage is distributed consistently and continually throughout the body. This explains why aging is spatially homogeneous and uniform and the declines in functions such as the maximum respiratory capacity, vital capacity, renal plasma flow, and glomerular filtration rate, all related to mitochondrial function and health, are clearly evident in aging [12].

## 3. Mitochondria and Metabolic Energy

At the biochemical level, metabolic energy is produced in semi-autonomous molecular units called respiratory complexes. The molecule that plays a central role in metabolism is ATP. It contains the stored metabolic energy in the form of a high-energy phosphate bond. This metabolic energy, which is released and made available by the breaking of this bond, ultimately allows every vital function of the cell to be carried out, including cellular damage repair. Only under proper energy conditions can life processes be maintained [16,17].

The mitochondrial production of ATP is based on the work of the four complexes (I–IV) of the electron transport chain, which gradually transfer electrons until oxygen is reduced to form water. The respiratory complexes that generate ATP are placed on rippled membranes inside the mitochondria, tiny organelles floating inside the cells. Each mitochondrion has five hundred to a thousand respiratory complexes, and each cell has between five hundred and a thousand mitochondria, depending on the type of cell and its energy needs. All cells convert energy in a similar way, through the hierarchy of respiratory complexes and mitochondria. Respiratory complexes are the critical elements of the network, and the optimal functioning of mitochondria occurs if the respiratory complexes act in a coordinated manner to provide energy to the cell [16,18]. Mitochondria are not only important for cellular bioenergetics but have recently been shown to be key regulators of stress responses and cell death [19]. These organelles, being responsible for various genetic diseases due to hereditary mutations of mitochondrial DNA (mtDNA), also playing key roles in the processes of inflammation, aging, carcinogenesis, and neurodegeneration and have attracted more interest in recent years in the field of biomedicine [20]. The regulatory role of mitochondria in intracellular metabolism is a potential and obvious link with senescence. Interestingly, in a screening for factors that could overcome senescence, it was found that the overexpression of the glycolytic enzyme cytosolic phosphoglycerate mutase was sufficient to bypass RAS-induced senescence [21]. Therefore, the balance between anaerobic cytosolic glycolysis and mitochondrial metabolism would seem to be an important modifier of whether a given senescent trigger induces stunting. Similarly, the direct inhibition of mitochondrial function, for example, by treating cells with rotenone or antimycin electron transport inhibitors, has been shown to be sufficient to cause senescence [22,23].

These observations place mitochondria both as effectors of downstream senescence and as potential initiators of upstream senescence. Interestingly, in the study in which electron transport inhibitors were able to induce senescence, it was also observed that a wide range of mitochondrial perturbations could trigger senescent cell arrest. 

## 4. Gerontogenes

The term “gerontogene” is used to refer to a genetic element that influences the lifespan of an organism by influencing the speed and/or onset of aging [24]. The story of the discovery of the genetic pathways of longevity (longevity pathways) began in 1988, when it was discovered, in the United States, that a worm, *Caenorhabditis elegans*, can live 65% longer if the gene “*AGE-1*” is eliminated from its DNA [25]. Seven years later, it was found that silencing another gene in a yeast model achieves the opposite effect, namely the shortening of lifespan. *SIRT* was the first gene of longevity to be discovered. Since 1988, about twenty gerontogenes and longevity genes have been identified in different species. Gerontogenes and longevity genes can also be found in mammals. It is hypothesized that, in humans, there are one hundred gerontogenes [26] *P66* is a gerontogene that plays an important role in mitochondria. The p66 protein acts in the mitochondria, where it accepts electrons and binds them to oxygen molecules to produce hydrogen peroxide (H_2_O_2_) [27]. Gerontogenes such as *P66* are the consequence of selection in the course of biological evolution in order to increase fat reserves and support cellular life in hostile environments with a scarcity of resources and low temperatures [28]. In 1999, it was discovered that the suppression of the *P66* gene in mice extends the lifespan by 30% [29]. Animal KO models for the *P66* gene are not only more long-lived but are also leaner and resistant to obesity. Furthermore, p66 influences energy metabolism, insulin sensitivity, and adipogenesis [30]. 

Finally, with the deletion of *P66* in mice, there is also a reduction in tumors that can be explained by a lower production of adipokines, molecules produced by adipose tissues that can drive tumor growth [31,32,33]. Mice without *P66* that are transferred to a stable in Siberia do not survive the cold temperatures, while those expressing p66 survive [34]. Ultimately, p66 is capable of increasing cellular resistance in hostile environments and leads damaged cells to apoptosis, a mechanism that ultimately allows the replacement of eliminated cells by stem cells. However, the pool of stem cells and the possibility that they will differ and replace the removed cells decrease with aging. The number of stem cells also decreases as a result of the arrest of the multiplication process of these cells because during each division telomeres, which have a protective function for the DNA, are shortened. When telomeres become too short, the cell can no longer divide and falls into a state of senescence. This leads to an increase in inflammation and an increased risk of tumors related to aging. This is linked to cellular DNA damage and reduced repair mechanisms. This is also due to the fact that p66 within the mitochondria helps to transform oxygen into hydrogen peroxide. H_2_O_2_ can cause reactions that lead to the formation of a hydroxyl radical, which is harmful to proteins and DNA [35,36].

Additionally, the enzyme superoxide dismutase (SOD) converts two molecules of superoxide—the product of reducing molecular oxygen with an electron—into one molecule of oxygen and one molecule of H_2_O_2_. The superoxide radical is known to be the most dangerous because it interacts with transition metals and H_2_O_2_. This leads to the production of hydroxyl, the most oxidizing radical found in nature, which is very aggressive towards protein structures and DNA [37,38].

Basically, SOD makes sure that superoxide and hydrogen peroxide are not present at the same time, thus producing an important antioxidant defense. However, H_2_O_2_ is also used for physiological processes by the cell, and in some cases its mitochondrial production acts as a messenger of the redox state [39], but it can also originate from NADPH oxidase and monoamine oxidase [40]. If the excess is not neutralized, it causes oxidative stress [41].

NADPH oxidase (NAO) is an enzyme, discovered by B. Babior, found on the outer membranes of macrophages and leukocytes. NAO produces superoxide and therefore other ROS, representing a defensive mechanism against bacteria or their components and constituting an important element of the innate immune response. NAO characterizes virtually all cells. The hydrogen peroxide produced by superoxide dismutation, catalyzed by SOD, plays a role in signal transduction pathways, similar to that produced in mitochondria [42,43]. Alongside *P66*, genes encoding peroxiredoxins (Prxs) have recently emerged as gerontogens linking aging to genome stability and nutrient-sensitive signaling. Prx deficiency causes accelerated aging in yeasts, worms, flies, and rodents [44,45,46,47]. Prxs act as scavengers of H_2_O_2_ and organic hydroperoxide and are catalytically inactivated during the aging process due to a modification caused by hydrogen peroxide itself [48]. The presence of gerontogenes explains the interaction between aging and the genome through the mechanisms of biological evolution. These genes are very active during an abundance of macronutrients and help to increase cellular metabolic activity to obtain as much energy as possible and store it as a caloric reserve in adipose tissue. The evolutionary purpose is twofold: to have an energy surplus available for reproduction and to accumulate energy in adipose tissues to ensure a reserve that can become indispensable in the case of starvation [49]. Humans had always dealt with periods with the availability of food and long periods of famine and fasting. The life of modern man has changed a lot, and we no longer live in primitive habitats. On the contrary, today we live in ecological niches with the continuous availability of food, but genes continue to be set to obtain as much energy as possible within the perspective of a potential paucity of nutrients. This behavior implies a shortening of life, resulting from the fact that the overproduction of energy in the cells takes place at the level of the mitochondria to obtain the primary source as ATP, but this process has a cost in terms of oxidative stress and cellular damage [50,51,52,53,54]. At the physiological level ROS act as redox messengers, while excesses damage cells [55]. ROS can play several physiological roles (i.e., cell signaling) and are normally generated as by-products of oxygen metabolism; despite this, environmental stressors (i.e., UV, ionizing radiations, pollutants, and heavy metals) and xenobiotics (i.e., antiblastic drugs) contribute to greatly increase ROS production, thereby causing the imbalance that leads to cell and tissue damage (oxidative stress). The idea that free radicals can cause aging dates back to the theory formulated by Denham Harman in the 1950s. However, the theory was wrong in the light of new knowledge [51]. Free radicals are not necessarily harmful and are rather useful for regulating mitochondrial respiration and the cellular response to danger [56]. In fact, free radicals, rather than attacking DNA and proteins randomly, are able to activate or deactivate certain key signaling proteins (including the mTOR protein, which regulates cell growth), which in turn regulate the activity of hundreds of genes and proteins. Consequently, we now know why high doses of antioxidants can eventually also produce harm and not just benefits [55,57], and oxidative damage is only one type of injury associated with aging [58]. In addition, programmed cell death (apoptosis), a process that is governed by mitochondria, is initiated by an increased release of free radicals from these organelles. Therefore, the production of some free radicals leads to apoptosis, which eliminates the cellular debris of old and dysfunctional cells. However, aged and dysfunctional mitochondria can block apoptosis, preventing this vital process, which is essential for cell renewal, and promoting the formation of senescent cells [59]. Harman’s theory needs to be revised into a new version that instead highlights the fact that the lifespan varies in almost all species in relation to the release of free radicals. The faster the escape of free radicals from the mitochondria to the cell, the shorter the lifespan. In general, the rate at which free radicals (FR) disperse depends on the metabolic rate, namely the rate at which cells consume oxygen [60,61]. If the escape of free radicals is fast from leaky mitochondria, degenerative diseases develop prematurely, while if it is slow, they occur later or do not emerge. It is common knowledge that lifespan reduction is inversely proportional to the cellular metabolic rate. The metabolic rate also decreases as the mass of an animal grows, so the damage per cell is reduced, with the result that larger animals live longer [62,63]. However, it is known that reducing the amount of food (i.e., eating less) is able to lower the cellular metabolic rate and FR release, thus potentially increasing the lifespan. Nevertheless, calorie restriction (CR) is still a controversial strategy to improve health. CR can both reduce inflammation and induce side effects such as body composition impairment, increased sensitivity to cold, reduced muscle strength, menstrual irregularities, infertility, the loss of libido, osteoporosis, slower wound healing, food obsession, irritability, and depression [64]. CR is supposed to exert its effects, at least in part, through metabolic pathways controlled by gerontogenes and longevity assurance genes [65]. If an animal is fed 30–40% less calories, it lives longer: 30% longer in mice and up to 200% longer in flies and spiders. Calorie restriction in animals not only increases the lifespan but also reduces diseases typical of aging, such as cancer and cardiovascular and neurodegenerative diseases [66]. A 30-year study showed how CR increases the lifespan by 30% in monkeys, halving the incidence of aging diseases such as cancers, cardiovascular diseases, diabetes, etc. [67,68]. It is believed that the effects of calorie restriction also apply to humans, but a 30% reduction in calories per lifetime in humans is difficult to achieve in normal socializing, where compared to laboratory experiments, there is no possibility of dosing food accurately, along with the risk of inducing malnutrition [69,70].

## 5. Calorie Restriction Mimetics and Senescence

Geroprotectors (GPs) are a set of substances that counteract aging, and they include antioxidants, hormones, immunomodulators, mimetics of caloric restriction, adaptogens, and others. GPs can have an outcome on aging similar to CR [71]. Thus, it is possible to have the benefits of CR without the drawbacks or dangers. Therefore, an easier way to take advantage of CR is the consumption of vegetables containing molecules that stimulate the same metabolic and molecular signaling pathways activated by CR, so called CR mimetics. These substances can be found naturally in several foods, such as red onions, capers, apples, extra-virgin olive oil, green tea, and others [72].

This strategy can lead to (1) the inhibition of gerontogenes and the activation of longevity genes, (2) the slowing of cellular aging through the activation of cellular stress response mechanisms, (3) a decline in cellular aging through the activation of molecular pathways and enzymes involved in age-related diseases such as type 2 diabetes and neurodegenerative and cardiovascular diseases, and (4) an improvement in mitochondrial function (Figure 1) [73,74,75]. Furthermore, aging is linked to an increase in the number of senescent cells (SCs). SCs are characterized by an irreversible replication arrest, resistance to apoptosis, and the frequent acquisition of a proinflammatory secretory phenotype associated with tissue senescence (SASP). SCs accumulate in different tissues with aging and characterize many chronic diseases and pathologic conditions. SASP may contribute to inflammation related to senescence, metabolic dysregulation, stem cell dysfunction, aging phenotypes, chronic diseases, geriatric syndromes, sarcopenia, fragility, and a loss of strength. SCs interfere with the normal metabolism and functioning of cells. Slowing the accumulation of senescent cells or reducing their presence is associated with the delay, prevention, or attenuation of multiple conditions associated with senescence [76,77]. Substances that clear SCs are called senolytics, and their benefits are under investigation [78].

## 6. Mitochondrial Aging and Natural Products with Protective Potential

Caloric restriction (CR) is widely studied for its ability to delay the onset of many disorders related to the aging process. CR is the most robust non-genetic intervention that reduces the rate of aging in mammals and other organisms and increases the lifespan [79]. The mechanisms responsible for the anti-aging effects of CR involve several processes, such as the activation of cell survival mechanisms. Among these, the reduced production of ROS by mitochondria is the main factor explaining the anti-aging effects of CR. Many experiments have established that CR stimulates mitochondrial biogenesis [79,80]. This, in the context of delayed mitochondrial aging, has generated growing interest in the scientific community. It was found that mitochondrial activity decreases with aging and that degeneration is linked to the reductions in mitochondria, mRNA transcripts, protein expression, and mtDNA and increased oxidative stress. In particular, reduced protein synthesis hinders protein turnover. The inability to replace damaged proteins may explain why aging is related to reduced mitochondrial function [79]. Mitochondria, the cellular organelles that produce most of the ATP, become dysfunctional during aging. This condition is coupled with inflammation, oxidative stress, and reduced cellular functionalities in every organ. Numerous genes have been linked to aging, with some favoring it (gerontogenes) and others blocking it (genes of longevity pathways). The desire for eternal youth is a constant in the history of humanity. The increased life expectancy in industrialized countries has unfortunately also led to a significant increase in the incidence of age-related diseases (ARDs) such as neurodegenerative diseases, diabetes, cardiovascular diseases, and cancers [81]. 

In this regard, caloric restriction (CR) can play a fundamental role by favoring the pathways of longevity, but being difficult to apply in a daily routine, it may be preferred to use CR mimetics, synthetic or natural compounds, to promote the same effects. These include certain phytonutrients such as oleuropein, epigallo-catechin-gallate, fisetin, and quercetin and minerals such as selenium that are able to exert their biological effects by influencing mitochondrial biogenesis, either directly by inhibiting specific enzymes or indirectly by modulating the signal to or from the mitochondria [20,82,83]. Some of them also have senolytic effects, which promote apoptosis in defective cells that accumulate over the years (senescent cells) and alter normal metabolism. Table 1 summarizes the main effects of natural vegetable compounds against aging and metabolic diseases.

### 6.1. Senolytic Effects of Oleuropeine and Hydroxytyrosol

Oleuropein (OLE) is one of the most abundant phenolic compounds extracted from olive oil and leaves, and it exerts anti-inflammatory and antioxidant effects related to more general cardio- and neuroprotective actions [83]. In vitro studies have shown anti-senescence effects that are mainly mediated by the induction of autophagy [84]. Oleuropein and its metabolite hydroxytyrosol (HT) have a powerful antioxidant activity, which could be responsible for the main antioxidant and anti-inflammatory activities associated with the use of olive oil [85]. The antioxidant activity of the phenolic compounds of olive oil has been studied through many experimental models. These compounds, including oleuropein, mainly tend to release hydrogen, forming intramolecular hydrogen ionic bonds between the free hydrogen compounds of the hydroxyl group and their radicals [108]. Moreover, they contribute to the regeneration of vitamin E and chelate iron ions, which in turn are able to initiate and propagate lipid pre-oxidation [109].

The consumption of extra-virgin olive oil is associated with reduced risks for most age-related diseases, including cardiovascular and neurodegenerative diseases (CVD and NDD) as well as some types of cancer [110]. Some of these effects are related to epigenetic mechanisms, especially histone modifications that modulate gene expression [110]. OLE has antioxidant, anti-inflammatory, antiatherogenic, hypoglycemic, lipid-lowering, and antiviral properties [83,84,85,108,109,110,111]. OLE also activates AMPK, a cellular sensor of the energy state that is linked to CR and is an activator of autophagy [86]. A portion of OLE is metabolized in humans as tyrosol and HT [87], substances with potential antiobesity effects [112]. There is extensive literature on HT on its multiple anti-aging and mitochondrial-function-enhancing effects [113,114]. Furthermore, OLE activates the proteasome system. This system is involved in many essential cellular functions, such as cell cycle regulation, cell differentiation, signal transduction pathways, antigen processing for appropriate immune responses, stress signaling, inflammatory responses, and apoptosis. Thanks to the proteasome, the cell periodically removes aberrant and defective proteins while avoiding the unfolded protein response (UPR), a cellular response related to the inability to degrade defective proteins, which leads to diseases typical of aging, such as neurodegenerative diseases [115,116]. The UPR and an altered proteasome characterize aging cells and chronic diseases [88,117].

### 6.2. Quercetin

Quercetin (QUE) is a natural flavonoid found in various vegetables. The foods richest in QUE are onions, apples, capers, blueberries, kale, chili peppers, tea, and broccoli. QUE from foods and supplements is bioavailable, in particular if consumed with fatty foods. The most absorbable form is the glucoside found in onions [118]. QUE is able to exert many actions on molecular signaling pathways by increasing the transcription of PGC-1α [89]. PGC-1α is a transcription coactivator of longevity-related genes that improve cellular antioxidant defenses by promoting mitochondrial activity and simultaneously lowering oxidative stress. In such a way, PGC-1α activation reduces inflammation and insulin resistance [119]. The reductions in inflammation and oxidative stress are able to lower senescence in experimental models [90]. QUE can modulate pathways associated with mitochondrial biogenesis, the mitochondrial membrane potential, oxidative respiration and ATP anabolism, and the intramitochondrial redox state; moreover, the apoptosis is correctly modulated [91]. Mitochondrial biogenesis is fundamental for healthy aging. Healthy immunosenescence allows the repair of cellular structures. Mitochondrial biogenesis can optimize cell function and survival in vitro and in vivo and determines cellular recovery from injuries caused by damaging environmental, pathophysiological, and/or infectious factors [120,121]. Moreover, QUE has other numerous beneficial effects on human health, acting as an anticarcinogen [92], an anti-infective [122], and a psychostimulant [123]. It also inhibits lipid peroxidation and platelet aggregation as well as the production of enzymes that produce inflammatory mediators such as cyclooxygenase (COX) and lipoxygenase (LOX) [124,125]. QUE also has a modulating and regulatory action on inflammation and immunity [126] and has shown protective effects against dexamethasone-induced skeletal muscle atrophy by regulating the Protein-Βx/Bcl-2 ratio and abnormal ΔΨm, leading to the suppression of apoptosis [127]. Moreover, QUE exerts positive metabolic effects on blood pressure, HDL cholesterol, and triglycerides [93] and pleiotropic effects on senescent cells [94]. Furthermore, QUE can have an inhibitory effect on adipogenesis [128]. In clinical trials, QUE is often combined with dasatinib, a tyrosine kinase inhibitor. This combination induces apoptosis in adipocytes [76] and increases the senolytic effect [90]. QUE, like other CR mimetics such as curcumin and EGCG, can protect the heart from aging and failure [95]. 

### 6.3. Epigallo-Catechin-Gallate

Epigallo-catechin-gallate (or EGCG) is a natural compound of plant origin belonging to the group of catechins, i.e., flavonoids from the family of polyphenols. Various types of tea contain ECGC, and the richest varieties are green tea and cocoa products [129]. Studies have reported that EGCG has many bioactivities, such as anti-inflammatory [96], anti-oxidant [97], antiviral [130], antimicrobial [131], antidiabetic [98], antiapoptotic, and anticarcinogenic properties [99], activities that underlie its role in cardiovascular and neurodegenerative diseases and metabolic syndromes. EGCG has anti-inflammatory effects by blocking proteins and factors that promote inflammation, in particular NF-kB, MAPKs, STAT, AP-1, and COX-2 [132]. In vitro and in vivo studies have shown that EGCG is able to decrease cytokine production, endothelial activation, and neutrophil migration in inflammatory disorders. Tea and ECGC may also reduce cardiovascular risk by improving endothelial function. Moreover, due to its chemical structure, it acts as a scavenger of free radicals and therefore has strong antioxidant properties, preventing the formation of reactive oxygen species and providing protection against oxidative damage [100].

EGCG can inhibit the progression of various types of tumors. It is claimed that EGCG, combined with other anticancer drugs (e.g., doxorubicin, cisplatin, and sunitinib) or other natural compounds (e.g., curcumin, ascorbic acid, quercetin, genestein, and caffeine) [133], has a synergistic effect for the treatment of hepatocellular carcinoma, breast cancer, and colon cancer [132].

It is also known as an important catechin with a neuroprotective effect due to its ability to maintain cellular homeostasis by modulating crucial intracellular signaling pathways implicated in the regulation of cell survival and apoptosis. Due to its ability to suppress the active oligomers of α-synuclein and amyloid aggregation, EGCG has been shown to be potentially prophylactic/therapeutic for Parkinson’s [134] and Alzheimer’s diseases, respectively [135]. In experimental models, EGCG increased the healthy lifespan in the worm *Caenorhabditis elegans*. The mechanism of the extension of life appears to work by stimulating the EGCG-induced production of ROS. In addition, EGCG activated mitochondrial biogenesis by restoring mitochondrial function. The increase in lifespan induced by EGCG depends on energy sensors such as AMPK/AAK-2, SIRT1/SIR-2.1, and FOXO/DAF-16 [136]. It is assumed that it is possible through the mechanism of para-hormesis, whereby a small dose of a substance stimulates the antioxidant defense mechanisms [137]. EGCG as well as curcumin and hydroxytyrosol activate the antioxidant response after being oxidized. This is due to their strong electrophilic nature, which is highly prone to oxidation [138]. After their oxidation, EGCGs activate the transcription of genes controlled by the ligands of an antioxidant response element (ARE). This, later renamed the “electrophile response element”, is triggered by a nuclear factor (Nrf2). Activated Nrf2 translocates from the cytosol to the nucleus. The stimulus translator is a protein named Keap1. Keap1 can induce degradation or translocation, depending on the electrophilic nature of the ligand via cysteine residues [139].

Many studies have fully demonstrated that EGCG has anti-inflammatory and antioxidant properties and improves lipid metabolism in animal experiments and human studies. EGCG increases longevity-related Sirt1 and FOXO1 protein expression by reducing oxidative stress and ROS generation. EGCG also influences the metabolism of fatty acids by inhibiting the activity of free fatty acid synthase (FAS) and acetyl-CoA carboxylase (ACC1) and the catalysis of palmite synthesis from acetyl-CoA and malonyl-CoA and by suppressing the synthesis of free fatty acid (FFA). Thus, the lifespan prolongation mechanism of EGCG-fed rats is related to metabolic effects [140,141]. The consumption of green tea is associated with a reduced risk of death from all causes. Tang et al. (2015) performed a meta-analysis of five studies including 200,884 subjects, concluding that drinking 2–3 cups (16–24 ounces) of green tea per day is associated with a maximum reduction in the risk of all-cause mortality of approximately 10% [142].

### 6.4. Fisetin

Fisetin (FIS) is a 3,7,3′,4′-tetrahydroxyflavone bioactive flavonol found in fruits and vegetables such as strawberries, apples, persimmons, grapes, onions, and cucumbers at concentrations between 2 and 160 μg/g. The average human daily intake is estimated to be around 0.4 mg [143]. FIS is considered to be an anti-inflammatory, hypolipidemic, hypoglycemic, antioxidant, neuroprotective, anti-angiogenic, and chemopreventive/chemotherapeutic agent [104]. Moreover, mitochondrial membrane depolarization and apoptotic cell death were reduced in aging rat brains treated with FIS through increases in the expression of autophagy genes and decreases in the expression of inflammatory genes [105]. Interestingly, it was shown that treatment with FIS diminished brain edema and deficit by decreasing the levels of proinflammatory cytokines. Moreover, a reduction in proinflammatory NF-κB signaling was evidenced after FIS treatment [144,145]. Intriguingly, an in vivo study on aging senescence-accelerated prone 8 (SAMP8) mice proved that FIS prevents cognitive and locomotor deficits. Additionally, three proteins linked to synaptic function were reduced in aged mice compared to young mice, and FIS treatment blocked their reduction almost completely [146]. Moreover, FIS significantly reduced ROS generation induced in mouse brains by D-galactose (a senescence accelerator), along with neuroinflammation-related pathways and pro-apoptotic markers [147]. Furthermore, FIS is considered an inducer of apoptosis, has a protective effect on the skin and the extracellular matrix, and increases the synthesis of collagen and the availability of glutathione, the main intracellular antioxidant in the human body [148,149]. FIS can protect LDL from oxidation and modulates SIRT-2, supporting cell repair after radical damage [150]. Recently, FIS has been recognized to have a senolytic activity, as it can eliminate senescent cells. A comparison between flavonoids revealed that fisetin was more effective than quercetin as a senolytic agent. FIS, unlike many others, can work as a single senolytic agent to counteract senescence, influencing lifespan and health [106]. FIS supplementation may provide neuroprotection against aging-induced oxidative stress, apoptotic cell death, neuroinflammation, and neurodegeneration in the rat brain. These physiological effects have been related to its ability to maintain the redox balance, ameliorated mitochondrial membrane depolarization, apoptotic cell death, and impairments in the activities of synaptosomal-membrane-bound ion transporters in the aging rat brain. FIS also acts by upregulating the expression of sirtuin-1 genes and autophagy genes (Atg-3 and Beclin-1) and downregulating the expression of inflammatory genes (IL-1β and TNF-α) and Sirt-2 in the aging brain [105]. In a rat model of accelerated senescence induced by D-galactose and in naturally aged rat erythrocytes, FIS supplementation significantly increases antioxidant levels and activates the plasma membrane redox system by suppressing aging-induced increases in ROS levels, eryptosis, lipid peroxidation, and protein oxidation [107]. Collectively, these effects favor correct cell functioning and slow aging. 

### 6.5. Curcumin

Turmeric (*Curcuma longa*), a rhizomatous herbaceous perennial herb, is a popular medicinal plant from Asia. Curcumin (CUR) is the main natural polyphenolic compound contained in turmeric, along with other secondary curcuminoids [151]. CUR is known for its effect against obesity and is particularly effective against ectopic fat. This is due to multiple mechanisms [152]. CUR’s effects on aging are progressively emerging. In Eastern populations with high consumptions of the spice curcuma, neurodegenerative diseases such as Alzheimer’s and Parkinson’s diseases have low incidences. As in the case of oleuropein and other molecules, this effect appears to be mediated by the action on the UPR, a mechanism that blocks the removal of aberrant proteins and is commonly found in diseases related to aging [101]. CUR modulates nutrient-sensing signaling pathways such as sirtuins and AMPK. Therefore, it is able to mimic caloric/diet restriction and increase the benefits linked to mild physical activity. CUR reduces the levels and activity of proteins involved in SASP and stimulates autophagy, favoring the renewal of cellular structures [153]. However, its role as a senolytic is not widely accepted [154]. Furthermore, curcumin can have an anti-aging effect via telomere protection; anti-inflammatory effects by the inhibition of NF-kB; and antitumor effects by modulating p53 [102]. The antioxidant effect of CUR is highlighted by the reduction in malondialdehyde, a marker of peroxidation, and the rise in the total antioxidant capacity, together with metal chelation and the augmented expression of enzymes related to ROS protection [103].

### 6.6. Senolytic Effect of Magnesium

In addition to the known effects of polyphenols, mineral salts also play a very important role in our health and in the prevention of cellular aging [155,156]. Magnesium, for example, plays an important role in many of the processes involved in regulating telomere structure, integrity, and cellular function. It is a divalent cation with a critical role in cellular metabolism [157] and is found in foods (whole grains, legumes, nuts, fruits, and vegetables) [158]. Water can also be a good source of magnesium [159]. The clinical relevance and biological significance of magnesium (Mg) has been documented in recent decades. Ferrè et al. demonstrated how Mg acts through the induction of the proinflammatory cytokine interleukin (IL)-1 alpha in cultured human endothelial cells. Indeed, the inhibition of IL-l alpha prevents the low-Mg-induced adhesion of monocytoid cells to the endothelium as well as the upregulation of the cdk inhibitor p21 [160]. Mg deficiency induces several characteristics typically associated with endothelial senescence. Mg deficiency, in addition to having a negative impact on the energy production pathway required by the mitochondria to generate ATP, also reduces the threshold antioxidant capacity of the aging organism and its resistance to free radical damage [161]. In fact, Mg also acts as an antioxidant against the damage of free radicals in the mitochondria [162]. Chronic inflammation and oxidative stress have both been identified as pathogenic factors in aging and various age-related diseases [81]. Mg deficiency over time causes the excessive production of oxygen free radicals and low-grade inflammation [162,163]. Despite the abundant distribution of magnesium in foods, several studies have indicated deficient intake, so much so that in American adults dietary magnesium intake is ∼70% lower than the reference dietary intake (DRI). As an essential mineral, the amount of magnesium within an organism must be continuously regulated, and distribution to individual cells must be ensured [164]. In the 1950s, the pathological focus of magnesium for various conditions in humans was introduced, and thereafter the importance of magnesium in physiological processes and in medicine was widely established [165]. The magnesium content in adult humans is about 0.4 g/kg, of which more than half is associated with bone connective tissue, while 38% is intracellular, mainly in plasma. Magnesium is mainly stored in bones but is also stored in striated muscle tissues, where it is associated with adenosine triphosphate, phospholipids, and proteins. In addition to carrying out structural functions, magnesium acts as a cofactor in about 300 enzymatic reactions, some of which are fundamental in glycolysis and the beta oxidation of fatty acids [164,166]. Magnesium in its ionic form (Mg^2+^) regulates various processes, including antioxidant and anti-inflammatory responses, and plays an important role in the proper functioning of other micronutrients, such as vitamin D [167]. Mg^2+^ participates as a second signaling messenger in the activation of T lymphocytes. Mg^2+^ deficiency can cause immunodeficiency, an exaggerated acute inflammatory response, a loss of antioxidant capacity, and an anti-inflammatory response by reducing the levels of nuclear factor kappa B (NF-κB), interleukin (IL)-6, and tumor necrosis factor alpha. Furthermore, supplementing Mg^2+^ improves mitochondrial function and increases the antioxidant glutathione (GSH) content, reducing OS [166,168]. Therefore, supplementing with Mg^2+^ is a potential way to reduce inflammation and OS while strengthening the immune system to manage COVID-19 [165,167]. These narrative reviews address the Mg^2+^ deficiency associated with worse disease prognosis, the supplementation of Mg^2+^ as a potent antioxidant, and anti-inflammatory therapy during and after COVID-19 and suggest that randomized controlled trials are needed. Studies show that chronic Mg deficiency can lead to increased oxidative stress and low-grade inflammation, which can be linked to various age-related diseases, including a greater predisposition to infectious disease. Hypomagnesemia is strongly related to oxidative stress markers, contributing to reductions in the expression and activity of antioxidant enzymes (glutathione peroxidase, superoxide dismutase, and catalase) and decreased concentrations of cellular and tissue antioxidants, in addition to increases in the production of hydrogen peroxide and superoxide anions by inflammatory cells [162]. Furthermore, an inadequate daily intake of magnesium can make an individual susceptible to infectious diseases. Alzheimer’s disease is one of the leading causes of dementia. This disease is the sixth leading cause of death in the United States, with over 79,000 deaths annually [169,170]. Some researchers have studied the magnesium balance in patients with mild to moderate Alzheimer’s disease. The study group included 101 older patients (73.4 ± 0.8 years of age; 42 men and 59 women) who were evaluated for total serum magnesium and ionized serum magnesium concentrations and underwent a Mini-Mental State Examination. This study showed that ionized magnesium concentrations were significantly related to cognitive function and not physical function, and individuals with Alzheimer’s disease had significantly lower Mini-Mental State Examination scores (20.5 ± 0.7 versus 27, 9 ± 0.2; *p* < 0.001) and significantly lower scores for physical function tests. This indicates that there is a correlation between the ionized magnesium concentrations and individuals with mild to moderate Alzheimer’s disease [171]. This knowledge shows us how magnesium can be considered a senolytic element, which is worth investigating in order to understand how it can be used to prevent or delay the processes of aging, an activity that magnesium could activate through a possible modulation of the SASP phenotype [172]. This could lead to new therapeutic strategies in humans towards related aging pathologies. Table 2 summarizes the main effects of magnesium. 

### 6.7. Selenium

Selenium (Se) is an essential trace element that was identified by Berzelius in 1817 and is involved in a multitude of cellular processes. Selenium was originally identified as a toxic element. However, in 1957, studies showed that selenium (along with vitamin E) was essential for the prevention of liver necrosis [178]. This led to the awareness that selenium deficiency was responsible for cell death in skeletal muscle cells, vascular smooth muscle cells, human uterine smooth muscle cells, and cardiomyocytes and was a contributing factor to Keshan disease in humans [179,180]. Although toxicity at higher levels is still a serious problem, appropriate amounts of this element are required for optimal human health [181]. Inorganic selenium is mostly stored in plants via the sulfur assimilation pathway, whereas animals and humans utilize these sources later as vegetables, meats, and dietary supplements [182].

In recent years, there has been an increasing interest in compounds containing selenium for their environmental, biological, and toxicological properties and especially for their various activities in the prevention and treatment of diseases, including cancer and infections [183]. Small amounts of selenium are protective against liver necrosis in vitamin-E-deficient rats [184]. Selenium deficiency has been associated with reduced immunity and chronic inflammation [185].

A significant amount of research conducted on cell cultures and animal models indicates that Se plays essential roles in regulating the migration, proliferation, differentiation, activation, and optimal functioning of immune cells, thus influencing innate immunity, the production of B-cell-dependent antibodies, and cell-mediated immunity [173]. Recent evidence on the roles of selenium and selenoproteins in the production of eicosanoids, derivatives of PUFAs with 20 atoms of carbon, which are involved in inflammatory responses, suggest that selenium supplementation could mitigate the dysfunctional inflammatory responses that contribute to the pathogenesis of many chronic health conditions [174,186]. Se has been shown to exert antioxidant and neuroprotective effects by modulating mitochondrial function and activating mitochondrial biogenesis [187]. Its biological function is achieved through the insertion of this trace element into a family of proteins known as selenoproteins. Among these, the glutathione peroxidase (GSH-Px) family, which includes six isoforms (GPX 1–6) that have selenocysteine on each subunit, is a family of selenium-dependent enzymes [182]. GPX is a component of the antioxidant glutathione pathways that detoxify lipid peroxides and provide protection to cellular and subcellular membranes against the reactive oxygen species (ROS) damage responsible for many diseases such as inflammation, anemia, cardiovascular disorders, and atherosclerosis [175]. Mammalian selenoproteins also include thioredoxin reductase (TR 1–3), iodothyronine deiodinases (D 1–3), selenophosphate synthetase (SPS2), methionine-R-sulfoxide reductase 1 (MsrB1), and several thioredoxin-like selenoproteins, some of which may act as safeguards against oxidant-induced toxicity in cells [176]. Normal cellular oxygen metabolism in aerobic organisms results in the production of ROS. The impairment of intracellular redox homeostasis leads to the condition of oxidative stress, which can damage biological macromolecules, with consequent alteration of the cellular functions and molecular mechanisms controlling cellular senescence [188]. In this context, cellular senescence represents the risk factor for several age-related diseases, including neurodegenerative, oncological, and cardiovascular diseases [189]. Therefore, the reduction in ROS and the related cellular damage is the primary objective of the prevention of age-related diseases. Aging leads to reduced cellular functioning and therefore reduced fitness as well as other effects. The protein, lipid, magnesium, phosphorus, selenium, and niacin intakes seem to promote a better quality of life [177]. Table 2 summarizes the main effects of Selenium. 

## 7. Concluding Remarks on the Senolytic Actions of Natural Products

The biologically active molecules present in foods or natural extracts interact with each other, and these interactions can increase their biological activity, creating a synergistic effect [190,191]. Usually, bioactive substances (such as polyphenols or others) are formed by organic molecules represented by heteroatoms, such as nitrogen, oxygen, and sulfur and a metal ion in the center that has a key role in the structure, in particular in the receptor interaction and in the rigidity of the structural conformation, which are probably the result of a long evolution from natural selection [192,193]. Furthermore, the association of several polyphenols can increase bioavailability and synergies. Some examples of this effect could be the synergy between quercetin and EGCG or the case of foods rich in apigenin and rutin that improve the absorption and activity of quercetin [193,194]. Plant compounds are metabolically neutralized by hepatic detoxification processes or by increasing their urinary excretion. These metabolic pathways can lead to reduced biological action. Mounting evidence suggests that the metabolism of certain phytochemicals may actually increase their biological activity, as seems to be the case for flavonoids. Previously, it was thought that the absorption of nutrient and non-nutrient molecules mainly took place in the upper small intestine and that the gut microbiota contributed to biotransformation, degradation (catabolism), and excretion. Some opinions in phytochemical research are now considering the impact of the microbiota on human health [101,195]. There are also additional confirmations indicating that phytochemicals, such as flavonoids and other polyphenols, may not even require absorption to exert their biological activities. Emerging evidence suggests that polyphenols affect the body directly in the gut through interactions with bacteria and human immune processes in the gastrointestinal tract and then other areas of the body [196]. Antioxidants from food and supplements can feed the microbiota and select health-related species [197].

In conclusion, a diet with an abundance of vegetables and sufficient variety, such as the Mediterranean diet, can ensure the supply of a series of phytonutrients that contribute to increasing life expectancy and health [198,199]. It is emerging from animal models and human trials how the supplementation of some of these nutrients can counteract mitochondrial dysfunction and promote the apoptosis of senescent cells, which are responsible for the alteration of the metabolism that is common in aging and can therefore help prevent and possibly manage the typical diseases of the elderly, such as tumors, CVD, and neurodegenerative diseases, and help in conditions such as obesity or metabolic surgery that may exhibit nutritional deficiencies or increased needs [200]. This seems to be linked, in particular, to an improvement in cellular energy production and a reduction in inflammatory processes [72,201,202,203,204]. A combination of the correct dose of physical activity [9,205], an appropriate gut microbiota [199], and genetic predisposition [206], together with an antioxidant-rich diet, is the key for successful aging. Cellular senescence has been shown to be a key mechanism driving aging, and CR can prevent senescent cell accumulation in both mice and humans. There is growing evidence that supplementation with some natural compounds can mimic CR without its side effects, protecting mitochondrial function and lowering inflammation. 

In this review, we have summarized the current knowledge on the cellular events associated with CR and discussed the potential roles of some natural senolytic agents in the treatment of age-related diseases.

## Figures and Tables

**Figure 1 ijms-23-16219-f001:**
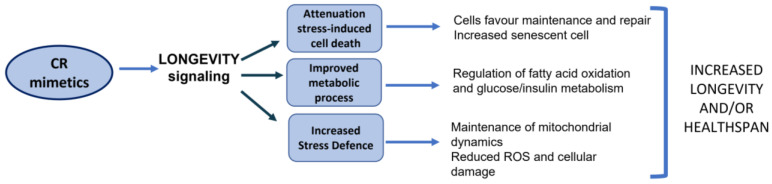
Mechanisms by which CR mimetics restrain the induction and propagation of cellular senescence. CR is a mild stress that provokes a survival response in the organism, which boosts resistance to stress and counteracts the causes of aging. CR mimetics reverse aging-derived effects by igniting numerous cellular mechanisms involved in the improvement of the lifespan. CR: calorie restriction; ROS: reactive oxygen species.

**Table 1 ijms-23-16219-t001:** Main effects of natural vegetable products against aging and metabolic diseases.

Compound	Main Features		Reference
OLE/HT	Antioxidative effect Anti-inflammatory effect Autophagy inducer	Proteasome activator UPR blocker	[83,84,85,86,87,88]
QUE	PGC-1α activator Senolytic effect Mitochondrial biogenesis stimulator	Anticancer effect Cardioprotective effect Anti-inflammatory effect	[89,90,91,92,93,94,95]
EGCG	Anticancer effect Anti-neurodegenerative effects Antidiabetic effect	Anti-inflammatory effect Antioxidative effect Immunomodulatory effect	[96,97,98,99,100]
CUR	Antioxidative effect Anti-neurodegenerative effects UPR blocker	Antiobesity effect Autophagy inducer	[101,102,103]
FIS	Anti-inflammatory effect Senolytic effect	Autophagy inducer Apoptosis inducer	[104,105,106,107]

**Table 2 ijms-23-16219-t002:** Main effects of mineral products against aging and metabolic diseases.

Compound	Main Features		References
Mg	Immunomodulatory effect Anti-neurodegenerative effects Mitochondrial protection	Antioxidative effect Cardioprotective effect Anti-inflammatory effect	[161,162,163,167,172]
Se	Immunomodulatory effect Mitochondrial biogenesis stimulator	Anti-inflammatory effect GPX stimulant	[173,174,175,176,177]

## Data Availability

Not applicable.

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
