# Peer review of "Mitochondrial Aging and Senolytic Natural Products with Protective Potential"

_ijms, 2022, doi:10.3390/ijms232416219_

Round 1

Reviewer 1 Report

The review article "Mitochondrial aging and senolytic natural products with protective potential" deepens the potential effect of natural substances that can promote a longer and healthier life. The sections are well presented and linked to each other. Interesting is the focus on phenolic compounds and minerals.  The manuscripts is well written and easy to read. 

Some minor modification are requested: 

-uniform the word epigallocatechin gallate 

-uniform in italics P66 (line 179)

- line 239, space before dot and It, plural in "calorie"

- line 255, correct the capital letter for "Improvement"

-line 288, insert the full name of the acronym EGCG

-line 313. repetition of line 299, autophagy 

-line 469-476, seems to fit better in the general intro of the paragraph 6

-line 497, dot missing after (DRI)

-line 582, delate comma

-line 620 and 629, space

-line 635-636, reformulate the sentence.

Author Response

Dear reviewer,

we resubmit our review with the changes made as a result of your revisions. Thanks for your suggestions which allowed us to improve our review "Mitochondrial aging and senolytic natural products with protective potential", detailed responses to your reviews are below.

I trust that you will find our revised version of ample interest to warrant publication in the pages of IJMS. The co-authors have read the revised review and approved its resubmission.

Sincerely yours,

Sara Franceschelli

Reviewer 1

The review article "Mitochondrial aging and senolytic natural products with protective potential" deepens the potential effect of natural substances that can promote a longer and healthier life. The sections are well presented and linked to each other. Interesting is the focus on phenolic compounds and minerals.  The manuscripts is well written and easy to read. 

Some minor modification are requested: 

-uniform the word epigallocatechin gallate

As requested, we standardized the word epigallocatechin gallate.

-uniform in italics P66 (line 179)

As requested, we uniform in Italics.

- line 239, space before dot and It, plural in "calorie"

- line 255, correct the capital letter for "Improvement"

As requested, we have corrected typos at line 239 and line 255

-line 288, insert the full name of the acronym EGCG

We have entered the full name of the acronym EPG

-line 313. repetition of line 299, autophagy

We have revised both sentences

-line 469-476, seems to fit better in the general intro of the paragraph 6

We accepted your suggestion and inserted the sentence in paragraph 6

-line 497, dot missing after (DRI); -line 582, delate comma; -line 620 and 629, space

As requested, we have corrected typos at lines 497, 582, 620 and 629

-line 635-636, reformulate the sentence

Thanks for the suggestion, we've rephrased the closing sentence as requested.

Reviewer 2 Report

The review article “Mitochondrial aging and senolytic natural products with protective potential” by Deledda et al., addresses an important and timely topic as aging is an intricate physiological process where major sub-cellular organs like mitochondria plays a substantial role. Therefore, it is imperative to study and explore natural products that have a beneficial impact on the gerontogenes and help improve the health and lifespan.

Overall, the review is well written and encompasses relevant literature. Specific comments are mentioned below.

1. One major concern that I have is there are no figures in the review. The authors have included the mechanism of action for most of the natural products described. It would be better to depict the same through representative images.

2. What was the basis of choosing these specific senolytic compound to add in the present review and exclude some of them for example Dasatinib.

3. There are discrepancies in the spacing between words throughout the manuscript, Check the same and correct accordingly.

4. Line 24- Replace the sentence “and a mineral such as magnesium and selenium” with “and minerals such as magnesium and selenium”.

5. Divide the introduction in two paragraphs.

6. Line 63- In the last few years, the discovery of natural products that affect mitochondrial function ………. innovative drugs” Add some of the commonly used/ studied natural products other than senolytics used as innovative drugs.

7. Line 248- Therefore, an easier way to take advantage of CR is the consumption of vegetables containing molecules ……………. by CR, so called CR mimetics” it would be better to add some of the vegetables/dietary products that are high in CR mimetics.

8. Table 1- Add references for all the compounds in the table in front of them. Do the same for Table-2.

9. There is a number 210 in the reference list but no reference, make changes accordingly.

Author Response

Dear reviewer,

We resubmit our review with the changes made as a result of your revisions. Their suggestions allowed us to improve our review "Mitochondrial aging and senolytic natural products with protective potential", detailed responses to our reviews are below.

I trust that you will find our revised version of ample interest to warrant publication in the pages of IJMS. The co-authors have read the revised review and approved its resubmission.

Sincerely yours,

Sara Franceschelli
